# Analysis of Morphological-Hemodynamic Risk Factors for Aneurysm Rupture Including a Newly Introduced Total Volume Ratio

**DOI:** 10.3390/jpm11080744

**Published:** 2021-07-29

**Authors:** Ui Yun Lee, Hyo Sung Kwak

**Affiliations:** 1Division of Mechanical Design Engineering, College of Engineering, Jeonbuk National University, Jeon-ju 54896, Korea; euiyun93@naver.com; 2Department of Radiology and Research Institute of Clinical Medicine of Jeonbuk National University, Biomedical Research Institute of Jeonbuk National University Hospital, Jeon-ju 54907, Korea

**Keywords:** cerebral aneurysm, computational fluid dynamics, hemodynamic, morphological, rupture

## Abstract

The purpose of this study was to evaluate morphological and hemodynamic factors, including the newly developed total volume ratio (TVR), in evaluating rupture risk of cerebral aneurysms using ≥7 mm sized aneurysms. Twenty-three aneurysms (11 unruptured and 12 ruptured) ≥ 7 mm were analyzed from 3-dimensional rotational cerebral angiography and computational fluid dynamics (CFD). Ten morphological and eleven hemodynamic factors of the aneurysms were qualitatively and quantitatively compared. Correlation analysis between morphological and hemodynamic factors was performed, and the relationship among the hemodynamic factors was analyzed. Morphological factors (ostium diameter, ostium area, aspect ratio, and bottleneck ratio) and hemodynamic factors (TVR, minimal wall shear stress of aneurysms, time-averaged wall shear stress of aneurysms, oscillatory shear index, relative residence time, low wall shear stress area, and ratio of low wall stress area) were statistically different between ruptured and unruptured aneurysms (*p* < 0.05). By simple regression analysis, the morphological factor aspect ratio and the hemodynamic factor TVR were significantly correlated (r^2^ = 0.602, *p* = 0.001). Ruptured aneurysms had complex and unstable flow. In ≥7 mm ruptured aneurysms, high aspect ratio, bottleneck ratio, complex flow, unstable flow, low TVR, wall shear stress at aneurysm, high oscillatory shear index, relative resistance time, low wall shear stress area, and ratio of low wall stress area were significant in determining the risk of aneurysm rupture.

## 1. Introduction

The size of cerebral aneurysms is correlated with the risk of rupture and is the most widely used determinant of treatment; aneurysms ≥7 mm are at increased risk of rupture [1,2]. However, according to the guidelines of the American Heart Association and American Stroke Association, morphological and hemodynamic factors, in addition to size, should be considered when estimating the risk of rupture [3]. Comparison of the features of ruptured aneurysms with those of unruptured aneurysms can help in identifying the risk factors for rupture and in the clinical management of the aneurysms [4,5].

Morphological factors, such as aspect ratio (AR) and flow angle of aneurysms are associated with rupture [6], and hemodynamic factors, identified with computational fluid dynamics (CFD) simulations, play a role in the pathogenesis, growth, and rupture of aneurysms. Among the hemodynamic factors are wall shear stress (WSS), oscillatory shear index (OSI), and relative residence time (RRT) [7,8]. These hemodynamic factors are used in the risk assessment and decision-making of aneurysm management [2,9], and potential additional factors should be considered.

The aim of the present study was to evaluate indicators to be used in determining the rupture risk of an aneurysm by validating the morphological and hemodynamic factors already used as well as a newly developed hemodynamic factor, total volume ratio (TVR), in unruptured and ruptured aneurysms ≥7 mm [10]. We quantitatively and qualitatively analyzed the morphological and hemodynamic factors and the correlations among the factors.

## 2. Materials and Methods

### 2.1. Patient Data

Patients with a cerebral aneurysm diagnosed during October 2015 and December 2018 at Jeonbuk National University Hospital were reviewed. The collection of patient data and the study design were approved by the hospital’s institutional review board (The Ethics Committee of Jeonbuk National University Hospital; JUH 2019-10-040). The acquisition time point of brain imaging of patients with ruptured aneurysms was as soon as the patients visited the emergency room. Inclusion criteria were as follows: (1) a source image performed with 3-dimensional rotational cerebral angiography; (2) saccular aneurysm shape; (3) aneurysm location (posterior communicating artery, distal internal carotid artery, cavernous, paraclinoid, and ophthalmic); (4) and aneurysm size ≥7 mm. Most patient data were excluded because of aneurysm size (mostly < 7 mm), aneurysm location, insufficient image quality to be used for CFD modeling, and the presence of fusiform or partially thrombosed aneurysms. Thus, 11 unruptured aneurysms and 12 ruptured aneurysms constituted the study population (posterior communicating artery, 17; distal internal carotid artery, 1; cavernous, 1; paraclinoid, 3; ophthalmic, 1). The age range of patients with unruptured aneurysms was between 40 and 80 years (62.81 ± 13.71 years), and those with ruptured aneurysms were between 41 and 86 years (62.16 ± 14.94 years).

### 2.2. Image Data Acquisition and Geometry Reconstruction

All image data were acquired with 3-dimensional rotational cerebral angiography (Axiom Artis dBA; Siemens Medical Solutions, Erlangen, Germany) with 1.5-degree rotation, 8-second-rotational image acquisition and 29 frames per second. Digital Imaging and Communications in Medicine(DICOM) format of the source image for each patient was obtained and imported into Materialise Mimics software (version 20.0; Materialise NV, Leuven, Belgium) for geometry reconstruction. Based on our previous work [11], we used threshold segmentation to obtain 3-dimensional geometry and a cropping method to designate the desired region of vessels. Images of small and unnecessary branches were removed and truncated for CFD analysis, and a smoothing method was used for rough and sharp surfaces of the reconstructed 3-dimensional geometry. The regenerated model was saved in a Standard Triangle Language format file for CFD simulation.

### 2.3. Measurement and Calculation of Morphological Variables

To quantitatively compare morphological variables between unruptured and ruptured aneurysms, height, width, size, ostium diameter (neck of aneurysm), ostium area, the surface area of the aneurysm, and volume of the aneurysm were measured. The largest distance between the tip of aneurysm domes and necks was defined as height, and the maximum diameter orthogonal to the height was defined as width. The size of aneurysms was defined as the longest diameter between height and width [12]. The maximum distance of the neck plane was defined as the ostium diameter [13]. The known morphological risk factors of aneurysm rupture, such as AR, bottleneck ratio (BR), and nonsphericity index (NSI), were calculated based on the measured morphological variables. AR was the ratio between the height of the aneurysm dome and the neck of the aneurysm dome, and BR was the ratio between the width of the aneurysm dome and the neck of the aneurysm dome [14]. NSI was calculated from the volume and surface area of the aneurysm dome [15]. The detailed equations are listed in Table 1 [6,15].

### 2.4. CFD Modeling

COMSOL Multiphysics Modeling software (version 5.2a; COMSOL Inc., Burlington, MA, USA) was used to generate mesh and solve the incompressible Navier–Stokes equation for computation of blood flow [16]. The following parameters of mesh element size were adjusted to find the optimal condition of mesh: maximum element growth rate, minimum element size, maximum element size, resolution of narrow regions, and curvature factor. The optimal condition of mesh was observed when no more changes in velocity and pressure were found. The mesh independence was confirmed. Tetrahedral meshes with approximately 500,000 elements were formed. Blood vessel walls were assumed to be rigid and to have a no-slip condition due to a lack of patient-specific information [17]. Assumptions of laminar flow, Newtonian fluid with a viscosity of 0.0035 Pa·s and density of 1066 kg/m^3^ were applied to the simulation [1,18]. The published flow rate (2.6 mL/s) and inlet area of each patient were used to calculate the patient-specific velocity profiles. The utilized flow rate for inlet boundary condition is shown in Figure 1 [8,17,18]. All simulations were computed during four cardiac cycles, and the second cardiac cycle was taken for quantitative and qualitative analysis [4].

### 2.5. Hemodynamic Data Analysis

After CFD simulation, the following hemodynamic variables were calculated for comparison of unruptured and ruptured aneurysms. (1) Time-averaged WSS (TAWSS) of the aneurysm was calculated by integration of the magnitude of the WSS during one cardiac cycle. The minimum and maximum WSS were also evaluated over the entire wall of the aneurysm dome. The time-averaged WSS of the parent artery was recorded for comparison with the aneurysm dome to the parent artery [4]. (2) OSI was defined as the fluctuation of WSS over one cardiac cycle. OSI is a non-dimensional parameter and ranges from 0 (no change) to 0.5 (oscillating flow) [19]. (3) RRT was calculated from WSS and OSI and defined as the residence time of blood near the aneurysm wall [6]. (4) Low wall shear stress area (LSA) was defined as the wall area of the aneurysm dome, indicating less than 10% WSS of the parent artery. The percentage ratio of LSA was calculated by dividing the LSA by the surface area of the aneurysm [20]. The detailed equations for TAWSS, OSI, RRT, and the ratio of LSA are listed in Table 1. (5) Flow complexity and stability: Flow complexity was classified as simple flow and complex flow. Simple flow had one recirculation zone, and complex flow had two or more recirculation zones. Stable flow and unstable flow were included in flow stability. Stable flow had no flow change during one cardiac cycle, whereas unstable flow had flow separation, change, and movement during one cardiac cycle [21]. 

### 2.6. Total Volume Ratio (TVR)

TVR is a recently developed quantitative hemodynamic factor that might analyze the rupture risk of an aneurysm. It is the ratio of blood volume through the aneurysm neck per one cardiac cycle to aneurysm volume.
(1)$$\mathrm{TVR}=\frac{\int_{T_{i}}^{T_{e}}Q_{AN}dt}{AV}=\frac{blood\;volume\;through\;aneurysm\;neck\;per\;one\;cardiac\;cycle}{aneurysm\;volume}$$

The volume flow rate of the blood through the aneurysm neck during one cardiac cycle is integrated and then divided by aneurysm volume for each patient. In Equation (1), $T_{i}$ and $T_{e}$ denote the initial time of the cardiac cycle and end time of the cardiac cycle, respectively. $Q_{AN}$ denotes blood volume through the aneurysm neck per one cardiac cycle, and *AV* denotes aneurysm volume. 

Simply expressed, high TVR means that blood flows well into the aneurysm and flows out well, whereas low TVR means that blood in the aneurysm is not fully circulating. For example, we assumed that aneurysm models A and B have the same aneurysm volume (250 mm^3^), and integrated blood volume flow rates of models A and B are 500 and 2500 mm^3^ per one cardiac cycle, respectively. Under these assumptions, the TVR of model A is $\frac{500\;mm^{3}}{250\;mm^{3}}$ = 2, and the TVR of model B is $\frac{2500\;mm^{3}}{250\;mm^{3}}$ = 10 (Figure 2). The low TVR in model A means that the ratio of the amount of circulating blood in the aneurysm to the aneurysm volume is five times lower than in model B. The cause of the low TVR is unstable and complex blood flow within the aneurysm, with blood stagnation. Low TVR is the comprehensive outcome of unstable, complex flow, low WSS, high OSI, and high RRT in the aneurysm. Thus, low TVR can be the combined results of morphological and hemodynamic variables.

### 2.7. Statistical Analyses

The measured and calculated morphological and hemodynamic variables were expressed as mean ± standard deviation. To compare values of unruptured and ruptured aneurysms, we used the Mann–Whitney test for non-normally distributed parameters and the independent sample t-test for normally distributed parameters. To analyze the relationships between morphological and hemodynamic factors, we used Pearson correlation analysis and simple linear regression. A *p*-value less than 0.05 was considered statistically significant. 

## 3. Results

### 3.1. Morphological Analysis of Unruptured and Ruptured Aneurysms

Comparison of measured and calculated morphological variables of unruptured and ruptured aneurysms are presented in Table 2. Ostium diameters of ruptured aneurysms were significantly less than those of unruptured aneurysms (6.08 ± 1.58 mm vs. 7.49 ± 1.40 mm; *p* = 0.035), as was ostium area (20.99 ± 11.51 vs. 33.92 ± 11.17; *p* = 0.013). The AR was significantly higher for ruptured aneurysms (1.32 ± 0.27) than for unruptured aneurysms (0.87 ± 0.26; *p* = 0.001). Similarly, BR was significantly more in ruptured aneurysms (1.62 ± 0.50) than in unruptured aneurysms ((1.18 ± 0.22), *p* = 0.009). Height, width, size, surface area, and volume of ruptured aneurysms and unruptured aneurysms were not significantly different. The NSI of unruptured aneurysms (0.29 ± 0.08) and ruptured aneurysms (0.30 ± 0.03) was very similar, and no significant difference was observed (*p* = 0.695). 

### 3.2. Quantitative Analysis of Hemodynamic Variables

The nine calculated hemodynamic factors of the two aneurysm groups are presented in Table 3. Values of the new hemodynamic factor, TVR, were significantly lower in ruptured aneurysms (5.02 ± 3.20) than in unruptured aneurysms ((10.62 ± 5.27), *p* = 0.005), and TVR had the lowest *p*-value among the calculated hemodynamic factors, which might allow for the most significant and representative results. TVR of unruptured aneurysms was two times larger than that of ruptured aneurysms. With respect to WSS, ruptured aneurysms had lower minimal WSS at the aneurysm, TAWSS at the aneurysm, and maximal WSS at the aneurysm than did unruptured aneurysms as shown in Table 3. Minimal WSS at aneurysms (*p* = 0.032) and TAWSS at aneurysms (*p* = 0.037) were significantly different between the two groups. TAWSS at the parent artery was higher than TAWSS at the aneurysm, and no statistically significant difference between TAWSS of the parent artery in unruptured and ruptured groups was found. Moreover, maximal WSS at the aneurysm was not significantly different between ruptured and unruptured aneurysms. Ruptured aneurysms had significantly higher OSI and RRT values compared to unruptured aneurysms (*p* = 0.031 and *p* = 0.006 for OSI and RRT, respectively). Similar to OSI and RRT, LSA and the ratio of LSA were significantly higher in ruptured aneurysms (55.92 ± 55.07 and 21.73 ± 9.79 for LSA and the ratio of LSA, respectively) compared with unruptured aneurysm (22.72 ± 21.95 and 12.71 ± 13.97 for LSA and ratio of LSA, respectively).

### 3.3. Qualitative Analysis of Hemodynamic Variables

Figure 3 shows the simulated flow streamlines of unruptured aneurysms at end-diastole. The streamline showed that all ruptured aneurysms had complex flow, whereas 72% of unruptured aneurysms had simple flow. Among the unruptured aneurysms, UR 6, 10, and 11 had complex flow; the remaining unruptured aneurysms had a single recirculation zone. In Figure 4, simulated flow streamlines of ruptured aneurysms are presented. All ruptured aneurysms had complex flow at end-diastole; the recirculation zones are indicated with black arrows. For example, RU 3 and 4 had two recirculation zones. RU 2 and 9 had one recirculation zone at the aneurysm tip. RU 1, 6, and 12 had multiple recirculation zones. The difference in flow complexity between the two aneurysm groups was statistically significant (*p* = 0.002) (Table 4).

Representative cases of unruptured aneurysms for flow stability and WSS are shown in Figure 5. UR 4 and 7 had no flow change during the cardiac cycle, and a small area of low WSS at the aneurysm domes was present. For comparison with ruptured aneurysms, scale bars of blood flow and WSS were the same (0 to 0.6 m/s for blood flow and 0 to 2 Pa for WSS). As presented in Figure 6, changes and movement of flow during the cardiac cycle were present in RU 1 and 3. Compared with RU 1 at peak-systole, blood flow at end-diastole became more complex. Similar to RU 1, RU 3 had unstable flow during the cardiac cycle. The changes in flow are marked with black arrows. Ruptured aneurysms had larger areas of low WSS at aneurysm domes compared to unruptured aneurysms. Unlike the WSS in unruptured aneurysms, because the region of low WSS area was large in ruptured aneurysms, a definite difference in WSS between the parent artery and the aneurysm dome was observed.

## 4. Discussion

Many studies have focused on finding risk factors of rupture of cerebral aneurysms by comparing morphological and hemodynamic factors between unruptured and ruptured aneurysms [2,4,8]. In this study, we analyzed 10 widely used morphological factors and 11 hemodynamic factors, including the new factor, TVR, for correlation with the rupture of aneurysms.

### 4.1. Morphological Variables

Ruptured aneurysms tended to have larger values of all factors except for neck diameter and area. The difference between unruptured and ruptured aneurysms was statistically significant only in neck diameter, neck area, AR, and BR (Table 2). Because the neck diameter of ruptured aneurysms was smaller than unruptured aneurysms, the values of AR and BR were increased. Moreover, ruptured aneurysms are known to have higher AR and BR, according to other studies [13,22]. 

AR is the most useful morphological factor in determining aneurysm rupture risk, so many researchers have conducted studies to set thresholds of AR [23]. For example, Ujiie et al. [24] studied 78 unruptured and 129 ruptured aneurysms and found that 90% of unruptured aneurysms had lower AR (less than 1.6), and 80% of ruptured aneurysms had greater AR (more than 1.6). However, Jing et al. [4] reviewed 86 unruptured and 69 ruptured aneurysms and found that AR of more than 1.064 was related to aneurysm rupture. Backes et al. [2] reported that, regardless of aneurysm size and location, AR more than 1.3 was associated with the risk of aneurysm rupture. In this study, we compared the average AR of unruptured and ruptured aneurysms and found that the AR of ruptured aneurysms was consistent with that reported by Backes et al. [2]. The average value of AR in ruptured aneurysms was 1.32, and the difference between the two aneurysms was statistically significant (Table 2). Although AR is a well-known morphological risk factor, reported threshold values vary among studies. In this study, the tendency of aspect ratio showed similar or different results compared to other literature. The results might be affected due to differences in the number of the study population, aneurysm location, and aneurysm size. Thus, validation of AR in large populations is needed. 

### 4.2. Hemodynamic Variables

In previous studies, there has been a consensus between in vivo measurement and CFD analysis, and hemodynamic factors can discriminate between unruptured and ruptured aneurysms. Thus, we calculated quantitative and qualitative hemodynamic factors using CFD simulation [8,25]. Complex flow, which increases infiltration of inflammatory cells from aneurysm walls, is associated with aneurysm rupture [4]. Cebral et al. [7] studied 210 patients with intracranial aneurysms and found that simple and stable flow were often present in unruptured aneurysms, whereas complex and unstable flow were correlated with aneurysm rupture. Similarly, in this study, most unruptured aneurysms had simple and stable flow, whereas ruptured aneurysms had complex and unstable flow significantly more often. 

Xiang et al. [8]. performed a quantitative hemodynamic analysis with 81 unruptured and 38 ruptured aneurysms. Xiang et al. found the difference between unruptured aneurysms and ruptured aneurysms. Ruptured aneurysms had lower WSS (0.68 Pa, unruptured vs. 0.33 Pa, ruptured), higher OSI (0.0035, unruptured vs. 0.016, ruptured), and higher RRT (2.70, unruptured vs. 7.52, ruptured) compared to unruptured aneurysms. In this study, we found that the ruptured group had lower WSS (1.17 Pa, unruptured vs. 0.73 Pa, ruptured), higher OSI (0.28, unruptured vs. 0.37, ruptured), and higher RRT (3.19, unruptured vs. 11.56, ruptured) compared to the unruptured group. The tendency of Xiang et al.’s study and this study was similar, but the values of hemodynamic factors were different due to the size variations. According to Xiang et al., the sizes of unruptured and ruptured aneurysms were 4.01 mm and 5.15 mm, respectively. However, we included only aneurysms larger than 7 mm in this study.

Chung et al. [1] compared unruptured and ruptured aneurysms with various hemodynamic factors. Lower minimal WSS (0.8 Pa, unruptured vs. 0.4 Pa, ruptured), higher LSA (45.8 mm^2^, unruptured vs. 51.6 mm^2^, ruptured), and higher OSI (0.25, unruptured vs. 0.31, ruptured) were found in ruptured aneurysms compared to unruptured aneurysms. We observed similar tendencies and values of hemodynamic factors compared to the Chung et al. study due to the similar aspect ratio of the aneurysm (0.879, unruptured vs. 1.304, ruptured in Chung et al.; 0.87, unruptured vs. 1.32, ruptured in this study). Jou et al. [26] conducted a comparison study between unruptured and ruptured aneurysms. The sizes of the unruptured aneurysms and ruptured aneurysms were 6.9 mm and 11 mm, respectively. The ratio of LSA was 11 for unruptured aneurysms and 27 for ruptured aneurysms. In this study, the ratio of LSA was 12.71 in the unruptured group and 21.73 in the ruptured group. Since this study included aneurysms larger than 7 mm, the tendencies and values of the ratio of LSA were similar to Jou et al.’s study. Minimal and time-averaged WSS at aneurysms were statistically significantly different between ruptured and unruptured aneurysms. However, maximal WSS at aneurysms and time-averaged WSS of parent arteries were not different. We only compared unruptured with ruptured aneurysms, but WSS was higher in parent arteries than in aneurysm domes because of the tortuosity of the vessels. A sudden change in morphology of blood vessels induces oscillating blood and high OSI in the aneurysm dome [27]. High OSI and low WSS promote damage to endothelial cells and degradation of the aneurysm wall, leading to aneurysm rupture [19]. Other studies have found that high OSI and low WSS are related to aneurysm rupture [28], and our results are consistent with those observations. Prolonged RRT and high OSI are associated with atherosclerotic changes in cerebral aneurysms [29]. 

Based on our observations on hemodynamic factors, the characteristics of ruptured aneurysms can help analyze the risk of rupture. Complex and unstable flow have multiple recirculation and oscillating zones. This feature causes decreased WSS and increased OSI. Due to the stagnation of blood flow, the region of low WSS is widened, and the residence time of blood flow near the aneurysm wall is longer. Thus, the LSA and RRT are increased. 

### 4.3. TVR

The newly introduced hemodynamic factor, TVR, is defined as the ratio of the integrated blood volume through the aneurysm neck per one cardiac cycle to aneurysm volume. We examined the correlation of TVR with morphological factor AR and hemodynamic factor TAWSS of aneurysms in Figure 2a and Figure 2b, respectively. In this study, ruptured aneurysms had complex, unstable flow, multiple recirculation zones, and stagnation of blood flow, which resulted in lower TVR of ruptured aneurysms compared to unruptured aneurysms. Low TVR denotes that circulating blood flow to aneurysm volume is relatively low, which means that recirculation zone and stagnation of blood are present in the aneurysm. Prolonged stay of blood flow promotes thrombus formation, and as thrombi get closer to the wall of the aneurysm, it promotes inflammation in the wall. The inflammatory reaction can lead to aneurysm rupture as various destructive enzymes are released [7]. 

As to the correlation between TVR and AR, due to the low TVR of ruptured aneurysms, low blood flow and stagnant flow in aneurysm domes are correlated with atherosclerosis and inflammatory reaction, resulting in the growth of aneurysms, and the growth might lead to an increase of AR [4]. In this study, TVR and AR were highly correlated (coefficient of determination: r^2^ = 0.602, *p* = 0.001). Thus, TVR might be a useful hemodynamic factor related to aneurysm rupture. For clinical use of TVR, calculation of TVR in more patient groups and various locations of aneurysms, such as the middle cerebral artery and anterior communicating artery, will be needed. 

### 4.4. Limitations of the Present Study

First of all, due to the small number of patients, there were several limitations as follows: (1) Small sample size limits the feasibility of statistical analysis (i.e., regression analysis); (2) This study focused on comparing unruptured and ruptured aneurysm groups rather than predicting rupture risk of the aneurysm due to the small number of patients; (3) The average values of the morphological factor (i.e., AR) and hemodynamic factor (i.e., TAWSS) could not be considered as a cutoff point due to the low number of cases. Studies in larger populations and multiple centers are needed to assess the validity of our results. Second, we conducted analysis on aneurysms larger than 7 mm. The results might limit the application of TVR in aneurysms smaller than 7 mm. Third, there might be errors in regenerating the geometry for each patient using the threshold segmentation method. However, since the images were acquired with 3-dimensional rotational cerebral angiography with high resolution, this might not be a significant limitation. Fourth, we assumed the vessels to have rigid walls and did not consider the real conditions of elasticity and thickness. Lastly, for CFD simulation, blood was assumed to be a Newtonian fluid since we did not have viscosity data to allow comparison of patients’ blood viscosity. The real physical property of blood is non-Newtonian, and blood flow becomes slower in aneurysm domes [30]. Ideally, we would have considered the non-Newtonian properties of blood, but the serious condition of patients with ruptured aneurysms was prohibitive. 

## 5. Conclusions

A detailed comparison of morphological and hemodynamic factors of modeled unruptured and ruptured aneurysms was conducted. In morphological analysis, large AR and BR were independently associated with rupture of aneurysms. In hemodynamic analysis, unstable flow, complex flow, low WSS, high OSI, RRT, LSA, and ratio of LSA were good indicators of rupture risk. A new hemodynamic factor, TVR, was significantly different between unruptured and ruptured aneurysms. These findings should be tested in multi-center, large population studies.

## Figures and Tables

**Figure 1 jpm-11-00744-f001:**
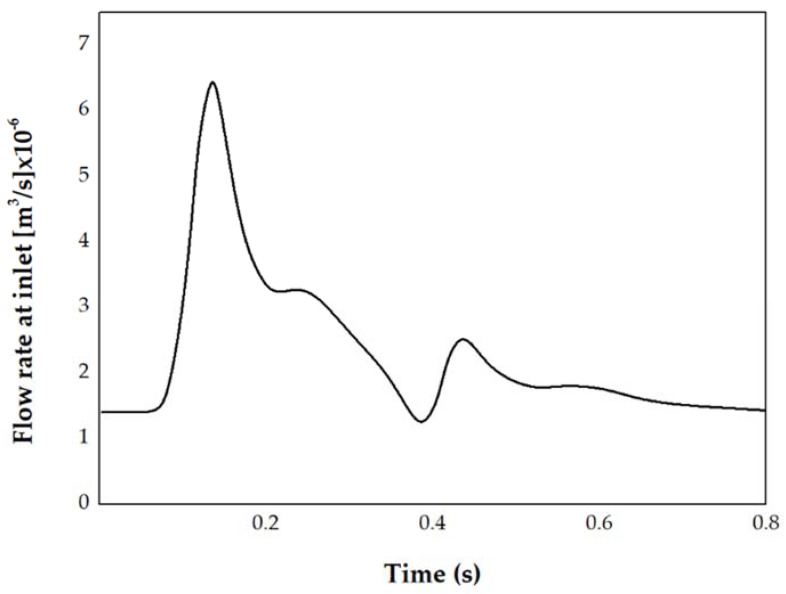
One cardiac cycle of flow rate according to time [18]. Using the flow rate and patient-specific inlet area, the velocity profiles for each patient were calculated for inlet boundary conditions.

**Figure 2 jpm-11-00744-f002:**
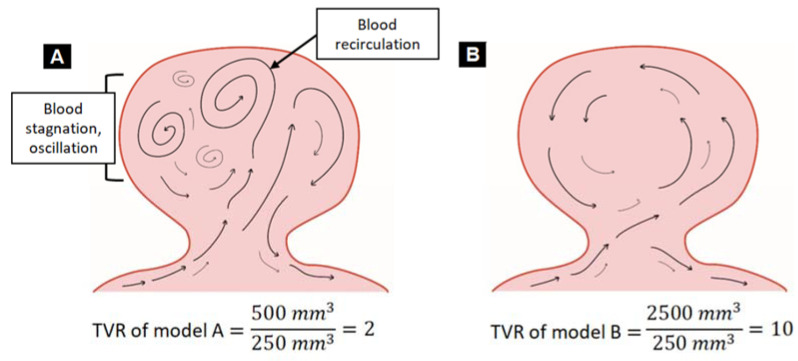
Two examples of a newly developed hemodynamic risk factor, total volume ratio (TVR). (**A**) TVR of model A shows lower TVR compared to model B due to blood recirculation, blood stagnation, and oscillation. (**B**) TVR of model B shows higher TVR compared to model A.

**Figure 3 jpm-11-00744-f003:**
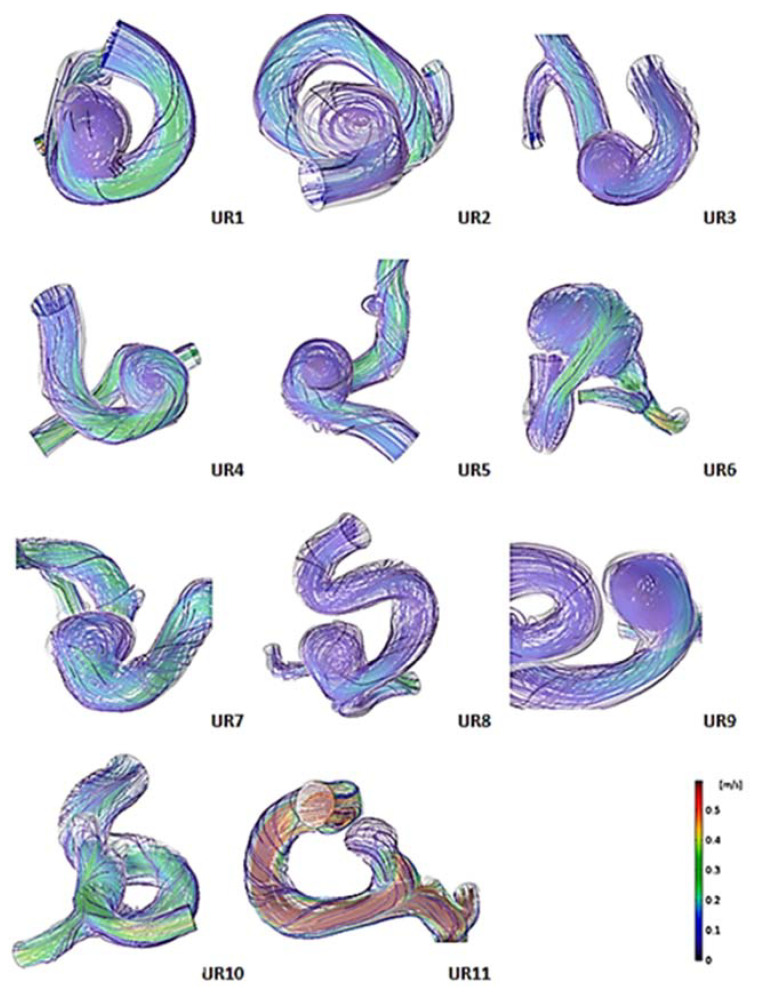
Simulated flow streamlines of unruptured aneurysms showing flow complexity at end-diastole. UR: unruptured aneurysm.

**Figure 4 jpm-11-00744-f004:**
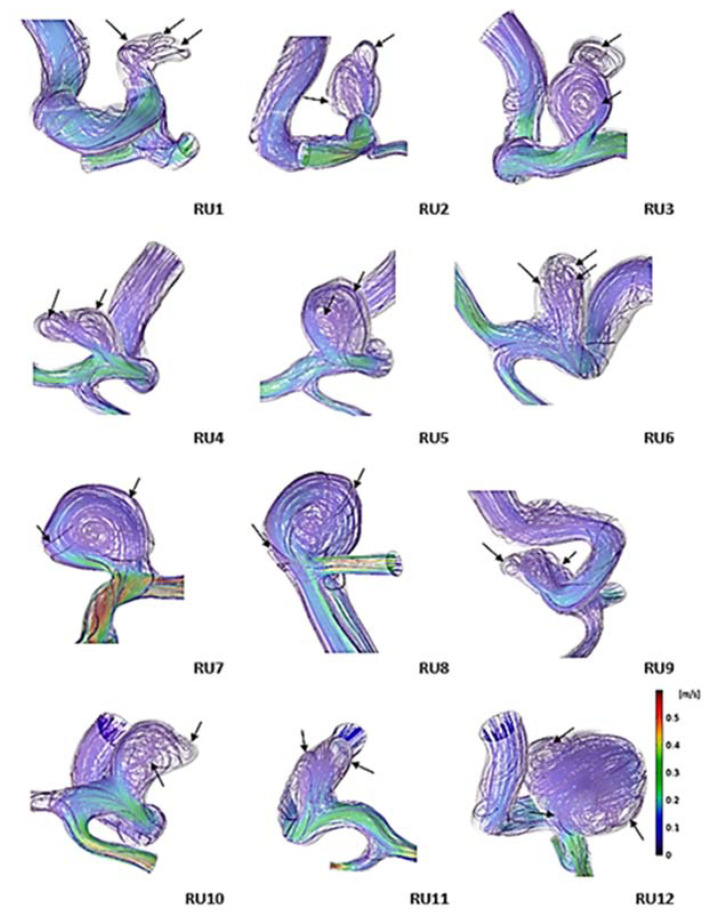
Simulated flow streamlines of ruptured aneurysms showing flow complexity at end-diastole. The black arrow indicates the recirculation region of the aneurysm. RU: ruptured aneurysm.

**Figure 5 jpm-11-00744-f005:**
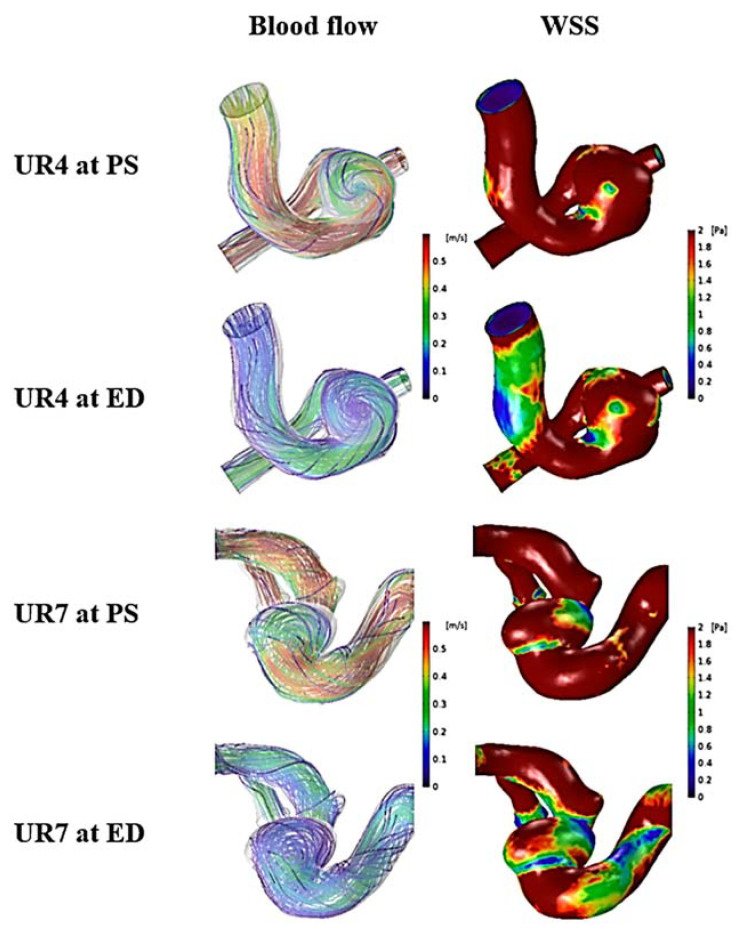
Two representative cases (UR4 and UR7) for unruptured aneurysms at peak-systole and end-diastole. The streamlines showing flow stability of the unruptured aneurysms (blood flow column). The distribution of wall shear stress on the aneurysm, showing low wall shear stress area (WSS column). PS: peak-systole; ED: end-diastole; UR: unruptured aneurysm; WSS: wall shear stress.

**Figure 6 jpm-11-00744-f006:**
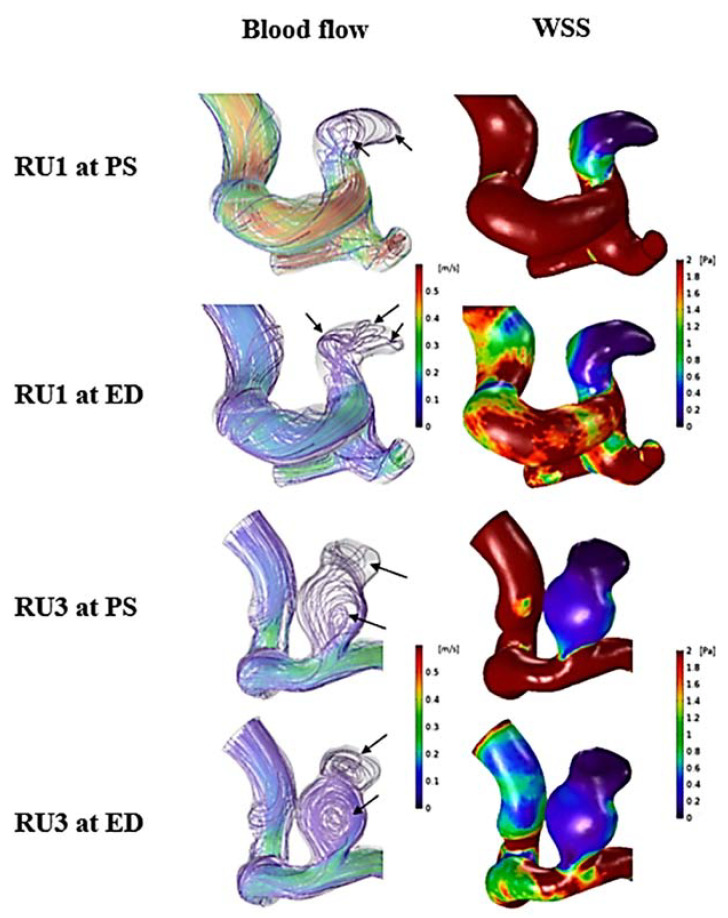
Two representative cases (RU1 and RU3) of ruptured aneurysms at peak-systole and end-diastole. The streamlines showing the flow stability of the ruptured aneurysm (blood flow column). The black arrow indicates the recirculation region of the aneurysm. The distribution of wall shear stress on the aneurysms showing low wall-shear stress area (WSS column). PS: peak-systole; ED: end-diastole; RU: ruptured aneurysm; WSS: wall shear stress.

**Table 1 jpm-11-00744-t001:** List of equations used for evaluation of morphologic and hemodynamic variables.

	Equations
**Morphological variables**	
AR	$\frac{\mathrm{height}\;\mathrm{of}\;\mathrm{aneurysm}\;\mathrm{dome}}{\mathrm{neck}\;\mathrm{of}\;\mathrm{aneurysm}\;\mathrm{dome}}$
BR	$\frac{\mathrm{width}\;\mathrm{of}\;\mathrm{aneurysm}\;\mathrm{dome}}{\mathrm{neck}\;\mathrm{of}\;\mathrm{aneurysm}\;\mathrm{dome}}$
NSI	1 − (18π)^1/3^$\frac{{\mathrm{volume}\;\mathrm{of}\;\mathrm{aneurysm}\;\mathrm{dome}}^{2/3}}{\mathrm{surface}\;\mathrm{area}\;\mathrm{of}\;\mathrm{aneurysm}\;\mathrm{dome}}$
**Hemodynamic variables**	
Time-averaged WSS	$\frac{1}{T}\int_{0}^{T}\left|WSS_{i}\right|dt$
OSI	$\frac{1}{2}\{1-\frac{\left|\int_{0}^{T}WSSdt\right|}{\int_{0}^{T}\left|WSS\right|dt}\}$
RRT	$\frac{1}{\left(1-2\times OSI\right)\times TAWSS}$
Ratio of LSA	$\frac{\mathrm{LSA}}{\mathrm{surface}\;\mathrm{area}\;\mathrm{of}\;\mathrm{aneurysm}\;\mathrm{dome}}\times 100$

Morphological variables: AR: aspect ratio; BR: bottleneck ratio; NSI: nonsphericity index. Hemodynamic variables: WSS: wall shear stress; TAWSS: time-averaged wall shear stress; OSI: oscillatory shear index; RRT: relative residence time; LSA: low wall shear stress area (area of low WSS below 10% of wall shear stress at parent artery).

**Table 2 jpm-11-00744-t002:** Quantification of morphological variables.

Variables (Unit)	Unruptured Aneurysm (*n* = 11)	Ruptured Aneurysm (*n* = 12)	*p* Value *
Patients			
Age (yr)	62.81 ± 13.71	62.16 ± 14.94	0.915
**Measured morphological variables**			
Height of aneurysm (mm)	6.45 ± 1.81	7.98 ± 2.54	0.115
Width of aneurysm (mm)	8.84 ± 2.01	9.54 ± 3.00	0.608
Size of aneurysm (mm)	9.10 ± 1.73	9.78 ± 2.93	0.512
Ostium diameter (mm)	7.49 ± 1.40	6.08 ± 1.58	0.035 *
**Calculated morphological variables**			
AR	0.87 ± 0.26	1.32 ± 0.27	0.001 *
BR	1.18 ±0.22	1.62 ± 0.50	0.009 *
Ostium area (mm^2^)	33.92 ± 11.17	20.99 ± 11.51	0.013 *
Surface area of aneurysm (mm^2^)	183.57 ±87.83	243.90 ± 175.91	0.260
Volume of aneurysm (mm^3^)	212.23 ± 175.70	361.32 ± 480.94	0.190
NSI	0.29 ± 0.08	0.30 ± 0.03	0.695

Values given are mean ± standard deviation, and * indicates *p* < 0.05. AR: aspect ratio, BR: bottleneck ratio, NSI: nonsphericity index

**Table 3 jpm-11-00744-t003:** Quantification of hemodynamic variables.

Variables (Unit)	Unruptured Aneurysm (*n* = 11)	Ruptured Aneurysm (*n* = 12)	*p* Value *
TVR	10.62 ± 5.27	5.02 ± 3.20	0.005 *
Minimal WSS at aneurysm (Pa)	0.69 ±0.37	0.42 ± 0.24	0.032 *
Time-averaged WSS at aneurysm (Pa)	1.17 ± 0.62	0.73 ± 0.42	0.037 *
Maximal WSS at aneurysm (Pa)	3.40 ± 2.17	2.14 ± 1.47	0.051
Time-averaged WSS at parent artery (Pa)	2.76 ± 1.08	3.30 ± 2.34	0.695
OSI	0.28 ± 0.11	0.37 ± 0.07	0.031 *
RRT	3.19 ± 2.00	11.56 ± 10.69	0.006 *
LSA (mm^2^)	22.72 ± 21.95	55.92 ± 55.07	0.037 *
Ratio of LSA (%)	12.71 ± 13.97	21.73 ± 9.79	0.023 *

Values given are mean ± standard deviation, and * indicates *p* < 0.05. TVR: total volume ratio, WSS: wall shear stress, OSI: oscillatory shear index, RRT: relative residence time, LSA: low wall shear stress area.

**Table 4 jpm-11-00744-t004:** Quantification of hemodynamic variables.

Complexity	Unruptured Aneurysm (*n* = 11)	Ruptured Aneurysm (*n* = 12)	*p* Value *
Simple	8	0	0.002 *
Complex	3	12	

* indicates *p* < 0.05.

## Data Availability

The data presented in this study are available on request from the corresponding author. The data are not publicly available due to privacy restrictions.

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
