# Peer review of "Analysis of Morphological-Hemodynamic Risk Factors for Aneurysm Rupture Including a Newly Introduced Total Volume Ratio"

_jpm, 2021, doi:10.3390/jpm11080744_

Round 1
Reviewer 1 Report
Please see the attached file.

Reviewer 2 Report
Manuscript revision.
In general it is correct, and the results are well presented.
However I have several observations before it can be published:
1. The input velocity profile must be reported.
2. The mesh used and a mesh independence study must be reported.
3. The relationships shown in Figure 2 do not seem adequate to me given the small number of samples, both broken and non-broken aneurysms follow the same trend, these graphs are, in my opinion, artificial and the linear regresion is not seen.
4. In the disusion, they should compare their morphological and hemodynamic values obtained with the literature and explain why they obtain different values from other studies, it is not enough to say that there are other studies that report WSS, AR, etc., the numerical values must be compared.
In summary, with so few cases, it is difficult to see the contribution, perhaps due to the new parameter that has a good P factor, it can be published, if the previously detailed observations are corrected, and an new manuscript is sended.
